# Sociotropy, Autonomy and Emotional Symptoms in Patients with Major Depression or Generalized Anxiety: The Mediating Role of Rumination and Immature Defenses

**DOI:** 10.3390/ijerph17165716

**Published:** 2020-08-07

**Authors:** Ruth Martínez, Carmen Senra, José Fernández-Rey, Hipólito Merino

**Affiliations:** 1Child and Youth Mental Health Unit, Álvaro Cunqueiro Hospital, 36213 Vigo, Spain; ruth.maria.martinez.barbosa@sergas.es; 2Department of Clinical Psychology and Psychobiology, Faculty of Psychology, University of Santiago de Compostela, 15782 Santiago de Compostela, Spain; jose.fernandez.rey@usc.es (J.F.-R.); hipolito.merino@usc.es (H.M.)

**Keywords:** sociotropy, autonomy, rumination, immature defenses, emotional symptoms, multiple mediation models

## Abstract

The relationships between dimensions of personality (sociotropy and autonomy), coping strategies (rumination: brooding and reflection subtypes, and immature defenses) and symptoms of depression and anxiety were explored in patients with Major Depressive Disorder (MDD) and Generalized Anxiety Disorder (GAD). A total of 279 patients completed questionnaires including measures of personality dimensions, rumination, immature defenses, depression and anxiety. Our findings suggested that sociotropy and autonomy may be associated with both depressive and anxious symptoms in patients with MDD and with GAD. Multiple mediation analyses indicated that brooding always acted as a mediating link between personality vulnerabilities (sociotropy and autonomy) and depressive and anxiety symptoms, independently of the patient group. In addition, in patients with MDD and those with GAD, brooding and immature defenses functioned together by linking sociotropy and autonomy, respectively, with depressive symptoms. Our results also showed that, in patients with GAD, both types of rumination explained the relationship between sociotropy and autonomy and anxiety symptoms. Overall, our findings provided evidence of the transdiagnostic role of the brooding, linking the vulnerability of personality dimensions and emotional symptoms. They also indicated that reflection and immature defenses can operate in conjunction with brooding, depending on the type of vulnerability and emotional context.

## 1. Introduction

It is known that the personality dimensions of sociotropy and autonomy can predispose individuals to the development of depressive symptoms [1,2]. Sociotropic individuals are characterized by an overvaluation of closeness and social acceptance in order to boost low self-esteem, while autonomous individuals base their self-esteem on achievement, independence and control. According to the vulnerability-stress hypothesis [1], the former, with an essentially interpersonal orientation, would be more vulnerable to depression when faced with events related to loss, criticism or abandonment, while the latter, with a more intrapersonal orientation, would have a greater risk of depression when faced with events that threaten their independence, achievements or control.

The relationship between these personality dimensions and depressive symptoms has been documented [3,4], although the results are more consistent for sociotropy than for autonomy [5,6,7,8,9]. Moreover, the role of these personality dimensions has also been investigated in relation to anxiety symptoms, although this evidence has been more limited and less conclusive. In this vein, Alford and Gerrity [10] found significant associations between sociotropy and symptoms of both anxiety and depression, but this result could not be replicated with that of autonomy. However, Fresco et al. [11] reported that sociotropy correlated with anxiety symptoms, while autonomy was associated with symptoms of depression. According to Sato, McCann and Ferguson [12], sociotropy was associated with the trait anxiety of social assessment, physical danger and ambiguous situations, while autonomy positively correlated with the trait anxiety of daily routines. Therefore, the question of whether these personality dimensions are associated or not with both emotional symptoms remains an open question, and it requires clarification [13]. 

As well as clarifying the nature of the relationship between personalities vulnerabilities and emotional symptoms, it is important to know how this connection occurs. That is, what variables could operate as an intermediate link by which certain personality vulnerabilities would lead to depressive and anxious symptoms (mediating hypothesis) [14]. It has been proposed that individuals, depending on their personality structure, will tend to prefer certain coping strategies for emotional regulation [15] and that such strategies may lead to different emotional symptoms [16]. 

Previous studies have largely investigated emotional regulation strategies that require conscious and deliberative resources, such as ruminative thinking. Rumination is an emotion-centered coping response, characterized by repetitive and unproductive thinking, through which individuals wrongly believe they achieve a better understanding of their depressive mood after an event [17]. Although the concept of rumination arises initially through being associated with depression [18,19], its contribution in the field of anxiety has also been confirmed [20,21,22,23,24]. However, there is very little literature examining the specific mediating role of rumination between personality dimensions and emotional symptoms. For example, Spasojević and Alloy [25] observed that rumination mediated the relationship between self-criticism and the subsequent development of depression, but this result could not be replicated with dependence (self-criticism and dependence are theoretically and operationally similar constructs to autonomy and sociotropy, respectively). The results of another study showed that avoidant coping does not directly mediate the relationship between sociotropy and symptoms of depression and anxiety [6]. Both studies involved non-clinical samples and analyzed rumination as a single construct. The ruminative process, though, has two different components that have been empirically established in both adolescents and adults: “brooding”, which refers to a passive, cyclical focus on negative emotions, and “reflection”, referring to emotionally neutral pondering [26]. While the maladaptive function of brooding has been widely documented [27,28,29], the quality of reflection is not yet clear [27,30].

Another psychological mechanism that could be involved in this relationship is that of defense styles [31,32]. Defense styles are also emotion-focused coping responses that often operate at an unconscious level, to minimize conflict or discomfort [33]. The Vaillant classification system [34] organizes defense styles according to different levels of maturity, so that mature defense facilitates good adjustment and mental health, whereas neurotic and immature defenses are usually maladaptive and distort the perception that individuals have of themselves or their surroundings. Both neurotic and immature defenses have been associated with depression [35,36,37,38] and anxiety [39,40]. Beyond such relationships, only one study [31] has investigated the mediating role of defense styles between personality dimensions (dependence and self-criticism) and emotional symptoms, finding that immature defenses only mediated the effect of dependence on depression. 

It is important to note that rumination and defense styles are not duplicate mechanisms of the same process, but two alternative forms of the general capacity of individuals to cope with a situation, and as such can provide a more detailed picture of the complex process of adaptation [37]. 

In sum, few studies have investigated the relationship between personality dimensions and emotional psychopathology by simultaneously analyzing more than one diagnostic category and more than one emotional regulation strategy as an intermediate link between both variables [16]. To address this gap in current knowledge, the present study proposes the following objectives:

(a) To determine the relationship between personality dimensions (sociotropy and autonomy) and the symptoms of anxiety and depression, respectively, in two groups of patients diagnosed with Major Depressive Disorder (MDD) or Generalized Anxiety Disorder (GAD), in order to know whether both dimensions are specific or transdiagnostic vulnerability factors.

(b) To explore the potential mediating effect of rumination (brooding and reflection) and immature defense styles between the personality dimensions and the symptoms of depression and anxiety, respectively, in order to determine whether these coping strategies operate in both types of patients or whether they function selectively.

Available research supports the notion that MDD and GAD are the most comorbid emotional disorders [41,42]. Furthermore, there is a high prevalence of anxiety symptoms in MDD and depressive symptoms in GAD that do not meet the diagnostic criteria for a full-blown disorder [43]. The inclusion of patients diagnosed with pure MDD or pure GAD allows us to know whether the use of coping strategies is specific or transdiagnostic in emotional disorders. Thus, the inclusion of these two groups of patients provides information when the symptoms evaluated as dependent variables are aligned with the main emotion (i.e., depressive symptoms in patients with MDD, or anxiety symptoms in patients with GAD) and when they are secondary symptoms (i.e., anxiety symptoms in patients with MDD, and depressive symptoms in patients with GAD).

## 2. Materials and Methods

### 2.1. Participants 

A total of 279 patients were recruited on an outpatient basis from the Psychiatric Services at the University Clinical Hospital of Santiago de Compostela (Spain), between March and December 2018. Those patients who met the criteria for diagnosis of Major Depressive Disorder or Generalized Anxiety Disorder were eligible. Patients with primary diagnoses other than MDD or GAD, and patients with any comorbid disorder, were excluded from the investigation, to control the coherence of the samples. A total of 119 patients met the criteria of diagnosis for MDD (86.6% female) and 160 patients for GAD (79.4% female).

### 2.2. Measures

The Personal Style Inventory-II (PSI-II). The PSI-II [4] is a 48-item measure of the two personality dimensions, sociotropy and autonomy (24 items on each scale). The sociotropy scale takes into account aspects related to pleasing others, dependency and concern over what others think. The autonomy scale includes elements such as perfectionism/self-criticism, need for control and defensive separation. All items are endorsed on a 1–6 scale, from “strongly disagree” to “strongly agree”. In this study, for the MDD group, Cronbach’s alpha was 0.86 for the sociotropy scale and 0.81 for the autonomy scale. For the GAD group, the internal consistency of the sociotropy scale was 0.88, and 70 for the autonomy scale. We used a Spanish translation that we developed through a back-translation procedure.

Ruminative Response Scale (RRS). The RRS [44] is a 22-item self-report for assessing ruminative coping responses to depressed mood. Treynor et al. [26] found that 12 items from the RRS overlapped with depressive symptoms, so that the resulting 10-item version was used. These ten items supported a two-factor model: brooding (5 items) and reflection (5 items). The Spanish translation of the RRS by Hervás [45], which has been found to show adequate psychometric properties, was used. Cronbach’s alpha scores for brooding and reflection were 0.75 and 0.70, respectively, for the MDD group, and were 0.74 and 0.69, respectively, for the GAD group.

The Defense Style Questionnaire-40 (DSQ-40). The DSQ-40 [46] provides a valid assessment for 20 individual defenses, which are grouped into three defense styles: mature defense style (8 items), neurotic defense style (8 items) and immature defense style (24 items). For the purposes of this study, only immature defenses were used, because they are the most common in patients with depression [43]. Neurotic and mature defenses were ruled out in this study, the former because the neuroticism dimension is intermediate in terms of level of maturity [47], and the latter because it was only a matter of examining the role of maladaptive defenses. We used a Spanish translation that we developed by using the back-translation procedure. The respective alpha coefficients were 0.63 and 0.73 for MDD and GAD groups, respectively.

Beck Depression Inventory-II (BDI-II). The BDI-II [48] is a 21-item self-report for measuring the severity of current depressive symptomatology. The Spanish version by Sanz et al. [49], which has been shown to exhibit high psychometric quality, was used. Cronbach’s alpha was 0.91 for the MDD group and 0.87 for the GAD group.

Beck Anxiety Inventory (BAI). The BAI [50,51] is a 21-item scale developed to assess the severity of anxiety symptoms. Respondents are asked to rate each item on a 4-point scale, ranging from 0 (not at all) to 3 (severely, can barely stand it). The Spanish version by Sanz and Navarro [52], which has been shown to exhibit high psychometric quality, was used. Cronbach’s alpha was 0.81 for the MDD group and 0.86 for the GAD group.

### 2.3. Procedure

Patients were consecutively admitted to the study if they met the inclusion criteria. The diagnoses were made by following the Structured Clinical Interview for DSM-IV-TR axis 1 diagnoses [53], which was administered by trained clinical psychologists familiar with the DSM-IV-TR classification and diagnostic criteria [54].

Before receiving any form of treatment, all patients completed the battery of self-report measures, under the supervision of a member of staff. Eleven of the patients were excluded from the study because they did not complete all of the questionnaires.

Approval for the study was granted by the Bioethics Committee at the University of Santiago de Compostela. Informed, written consent was obtained from the patients after being provided with a full description of the study.

### 2.4. Analytic Strategy

Descriptive statistics and Pearson correlations were conducted. Multiple mediation analyses were performed with the PROCESS macro for SPSS [55]. Figure 1 illustrates the design of a multiple mediation model. A set of multiple mediation models were performed to assess whether coping responses—brooding, reflection and immature defenses (mediator variable = M)—have an indirect effect on the relationship between the dimensions of personality, sociotropy and autonomy, (independent variable = X) and depressive and anxiety symptoms (dependent variable = Y), respectively. Gender and age were controlled in these analyses. Pairwise contrasts of all the specific indirect effects involved in each model were also calculated as a mean of comparing the strength of the individual indirect effects of each mediator included in the model. Bootstrap resampling techniques (with 20,000 resamples) were performed, and a 95% bias correction confidence interval (BootCI) was used to generate confidence intervals for the hypotheses tested, on the basis that it is the preferred method for assessing indirect effects in both simple [56] and multiple mediator models [57]. A total of four models were tested for each group of patients.

## 3. Results

### Descriptive Analysis

The mean age of the MDD group was 43.2 years (SD = 11.5), and for the GAD group, it was 33.8 years (SD = 9.6). The distribution of marital status was similar in the two samples; approximately 45% were married, 40% were single, 13% were separated or divorced and 2% were widowed. In terms of educational level, the distribution was also very similar in the two groups: 32% had a low level of education, 35% had an average level, and 33% had a high level.

As a preliminary step, the data were analyzed to determine whether they met all the statistical assumptions [58]. Assumptions of linearity, normality and homogeneity of variance were fulfilled for all measures.

Descriptive statistics and Pearson correlations between the different target variables in each diagnostic group are shown in Table 1. As can be seen, both personality dimensions are positively and significantly associated in the case of patients with GAD; however, in the group of patients with MDD, this association was not significant. In both patient groups, coping strategies correlated significantly with depressive and anxious symptoms and with personality dimensions, except in the case of sociotropy and immature defenses in the MDD group.

The results of multiple mediation analyses were organized by comparing the two groups of patients according to the dependent variables: depressive or anxious symptoms. Thus, the results of the analyses concerning depressive symptoms for patients with MDD are summarized in Table 2, and for patients with GAD, in Table 3. In patients with MDD, the total indirect effect of sociotropy on depressive symptoms through all the proposed mediators was significant (i.e., the bias-corrected 95% confidence interval for the total indirect effect did not contain zero). Specifically, the mediators reduced the non-standardized regression coefficient of sociotropy on depressive symptoms from 0.22 (*p* ≤ 0.001) to 0.09 (*p* = 0.13), which reflected 59% [(0.22 − 0.09)/0.22] of the association between this personality trait and depressive symptoms. When examining the specific indirect effect of each mediator, only brooding presented a significant value. The pairwise contrasts between mediators were not significantly different from each other. This analysis was repeated on the basis of the trait of autonomy as an independent variable (see lower part of Table 2). The results indicated that the total indirect effect of autonomy by the potential mediators (brooding, reflection and immature defenses) on depressive symptoms was significant. The mediators reduced the non-standardized regression coefficient from 0.22 (*p* ≤ 0.001) to 0.02 (*p* = 0.75), accounting for 90% [(0.22 − 0.02)/0.22] of the association between independent and dependent variables. The specific indirect effects of brooding and immature defenses were significant. The pairwise contrasts revealed that the specific indirect effect of autonomy on depressive symptoms through brooding and immature defenses was not significantly different between them, but was different from reflection.

The results for the GAD group are presented in Table 3. The analysis concerning depressive symptoms from the sociotropy trait showed that the total indirect effect of the three proposed mediators was statistically significant and reduced the non-standardized regression coefficient of sociotropy on depressive symptoms from 0.24 (*p* ≤ 0.001) to 0.05 (*p* = 0.41), and explained 79.16% [(0.24 − 0.05)/0.24] of the association between the two variables. Exploration of the specific indirect effect of each mediator revealed that brooding and immature defenses were significant mediators, whereas the indirect effect of reflection was not significantly different from zero. The pairwise contrasts showed that the specific indirect effects of immature defenses were not significantly different from two components of rumination, but brooding was greater than reflection. With regard to the autonomy trait (see the lower part of Table 3), the total indirect effect of the three proposed mediators was found to be significant and involved a reduction in the non-standardized regression coefficient of autonomy on depressive symptoms from 0.31 (*p* ≤ 0.001) to 0.17 (*p* = 0.016), with this accounting for 45.16% [(0.33 − 0.17)/0.33] of the association between the two variables. Among the specific indirect effects within this analysis, only significant mediation was found through brooding. The pairwise contrasts revealed that the specific indirect effect of autonomy on depressive symptoms through brooding was not significantly greater than that of the other two mediators.

The results for the MDD in which sociotropy or autonomy were the independent variables and the anxiety symptoms were the dependent variable are presented in Table 4. The total indirect effect of sociotropy on anxiety symptoms through the proposed mediators (brooding, reflection and immature defenses) was significant. The mediators reduced the non-standardized regression coefficient of sociotropy on anxiety symptoms from 0.27 (*p* ≤ 0.001) to 0.11 (*p* ≤ 0.17), reflecting 59.2% [(0.27 − 0.11)/0.27] of the association between the independent and dependent variables. The examination of the specific indirect effect of each mediator showed that only brooding was significant. The pairwise contrasts revealed that the indirect effect of the brooding was greater than those for reflection and for immature defenses. This analysis was repeated with the trait of autonomy as an independent variable. The results indicated that the joint indirect effect of the mediators was significant, accounting for 54% [(0.22 − 0.1)/0.22] of the association between autonomy and anxiety symptoms. Furthermore, testing of specific indirect effects revealed that only brooding was a significant mediator. The pairwise contrasts showed that the indirect effect of brooding was only greater than that of reflection.

The results for the mediation analyses for the GAD patient group are summarized in Table 5. The total indirect effect of the three proposed mediators was statistically significant and reduced the non-standardized regression coefficient of sociotropy on anxiety symptoms from 0.25 (*p* ≤ 0.000) to 0.12 (*p* = 0.08), thus explaining 52% [(0.25 − 0.01)/0.25] of the association between both variables. When analyzing the specific indirect effect of the mediator, it was found that the two subtypes of rumination significantly mediated the relationship between sociotropy and anxiety symptoms. Moreover, the pairwise contrasts indicated that there were no differences in the magnitude of the mediators analyzed. Finally, regarding the trait of autonomy (see lower part of Table 5), the total indirect effect of the three proposed mediators was statistically significant and reduced the non-standardized regression coefficient of autonomy on anxiety symptoms from 0.15 (*p* ≤ 0.05) to 0.01 (*p* = 0.85), accounting for 93.3% [(0.15 − 0.01)/0.15] of the association between both variables. The individual contribution of each particular mediator indicated that brooding and reflection mediated significantly between variables. The pairwise contrasts revealed that there were no differences in the magnitude of the mediators analyzed.

## 4. Discussion

The purposes of the present study were (1) to investigate whether the personality dimensions of sociotropy and autonomy are specifically or transdiagnostically associated with depressive and anxiety symptoms and (2) examine the mediational role of rumination (brooding and reflection) and immature defense styles in the relationships between personality dimensions and depressive and anxious symptoms, respectively, in two clinical samples of depressed (MDD) and anxious (GAD) patients.

According to the first objective, we found that both dimensions of personality are significantly associated with depressive and anxious symptoms, regardless of the diagnostic group. Therefore, although such personality configurations were originally proposed as risk factors for depression [1], the current study, in line with other studies [59,60], showed that they are also related to anxiety symptoms. That is, both individuals whose personality configuration is characterized by an intense fear of not being loved and abandoned, and those who fear failure and loss of control, are vulnerable to developing emotional symptoms, which suggests that both personality dimensions can operate as transdiagnostic configurations of vulnerability [61]. Furthermore, it should be noted that, while in the MDD patient group, sociotropy and autonomy were shown to be independent constructs, in the GAD patient group, they were significantly related. This finding seems to suggest that the “modes of functioning, values and goals” that constitute vulnerabilities to emotional symptoms are more clearly delineated in MDD patients than in GAD patients. That is, in depressed patients the vulnerability was focused toward intrapersonal or interpersonal domain stressors; in patients with GAD, by contrast, the significant interrelationship between sociotropy and autonomy reflects the existence of a “shared feature” between both types of vulnerability.

The second aim of the study was to explore the mediating role of different coping strategies in the relationship between sociotropy and autonomy and depressive and anxious symptoms, respectively, in patients with MDD and with GAD. The examination of depressive symptoms as a dependent variable showed the existence of a pattern of reverse mediation between personality dimensions, when comparing both groups of patients. Thus, while patients with MDD used brooding and immature defenses jointly to cope with intrapersonal stressors, yet only brooding in the face of interpersonal threats, patients with GAD used brooding and immature defenses to deal with interpersonal domain stressors and brooding in the face of negative intrapersonal situations.

A first examination of these results, in line with previous studies, indicates that rumination is linked to maladaptive cognitive styles [62,63] and predicts depression beyond these negative cognitive styles [25]; in particular, the brooding rumination subtype has shown the most consistent and robust association with depression [27,28,29,64]. In this sense, the present study revealed that brooding is the link between personality vulnerabilities and depressive symptoms in patients with MDD and with GAD, thus demonstrating its transdiagnostic function. Furthermore, patients with MDD and intrapersonal vulnerability and patients with GAD and reactivity to interpersonal stressors activate not only conscious strategies such as brooding, but also unconscious ones like immature defense mechanisms. This finding seems to indicate, on the one hand, that, in such situations, individuals perceive a greater intensity of threat and, on the other hand, that both types of strategies have independent effects on depression [65]. In an earlier study with a community sample, it was reported that immature defenses mediated the relationship between dependent personality style and depressive symptoms [31]. The present study extends existing knowledge to a clinical sample and finds that immature defenses also mediate between autonomous personality and depressive symptoms. However, this result requires some clarification. In the group with GAD, immature defenses link sociotropic vulnerability with depressive symptoms; by contrast, in the group of patients with MDD, immature defenses mediate between autonomous personality and depressive symptoms. It should be added that, in both cases, the immature defenses act together with brooding. These results are only partly consistent with those reported by Calati et al. [66] and Colovic et al. [43], who found that patients with depression use immature defense mechanisms more often than patients with anxiety. In our study, it was observed that, regardless of the diagnostic group, MDD or GAD, immature defenses were always associated with depressive symptoms.

The examination of the anxious symptoms as a dependent variable showed that patients with MDD and with GAD use the same coping strategies regardless of their sociotropic or autonomous vulnerability. In particular, in the case of patients with MDD, only brooding mediated the relationship between the two dimensions of personality and anxiety symptoms. This result indicates that brooding, which was traditionally linked with depressive symptoms, is also associated with anxious symptoms [67,68]. In patients with GAD, in addition to brooding, reflection also intervenes, so that both subtypes of rumination link personality dimensions and anxiety symptoms. This finding confirms the maladaptive function of the reflection subtype of rumination [69,70]. Furthermore, the finding seems to reflect the fact that rumination subtypes can be differentially used, depending on the emotional context in which they occur [71], since when we examine anxiety symptoms in patients whose main emotion is anxiety (GAD), both rumination subtypes operate independently. It could be assumed that reflection may be contributing a “more active” quality to the perseverative process that overlaps with other repetitive cognitive styles commonly associated with anxiety symptoms, such as worry [63], a hallmark of generalized anxiety disorder [72]. In this regard, individuals with GAD might be considered to ruminate on problems that concern them too much [73].

This study has some limitations that may provide new opportunities for future research. First, the cross-sectional design of the investigation precludes conclusions being made about causality. Future studies should examine these mediational models longitudinally. Second, only self-report measures were used, which may have introduced some bias in the results, particularly in the case of the DSQ. Although this is the most widely used self-report measure of defense mechanisms [74], its validity has been questioned because defense mechanisms are postulated to operate outside of conscious awareness [75]. Nevertheless, the DSQ essentially appraises the conscious manifestations of defensive activity [37], and the habitual use of certain defense styles probably results in conscious derivatives that may be accessible to the individual [45]. Third, the current sample consisted solely of Caucasians, so the results might be interpreted with some caution with regard to any generalizability to other, broader samples. Fourth, the results cannot be generalized to other disorders; however, it would be interesting to test the preferential use of coping strategies in other clinical groups.

Finally, at the practical level, the present study emphasizes the importance of rumination and immature defenses as carriers of vulnerability in sociotropic and autonomic personalities. Therefore, Rumination-Focused Cognitive Behavior Therapy [76] would enable patients to develop new strategies to meet challenges and to replace the abstract, self-referential loop characteristic of rumination for a constructive style of thinking, through the use of functional analysis and behavioral experiments, for example. Furthermore, it would be beneficial to incorporate elements of metacognitive therapy [77] as a means of addressing the underlying cognitive process that exacerbates and maintains these rigid and perseverative thought chains (i.e., attention training and postpone rumination). In addition, given that immature defenses appear to interact with brooding in both MDD and GAD patients, which increases vulnerability to depressive symptoms, treatment should also involve addressing and working on defense styles, in an attempt to make patients fully conscious of them, and thus to facilitate the modification of these. Some authors consider that, rather than acting directly on immature defenses, which may lead to negative reactions, training in the development of flexible coping skills is preferable [78,79].

From the perspective of prevention, given that ruminative thinking style often exhibits a predictable trajectory [80], it would be useful to train individuals in a more flexible and productive thinking style, one that provides a greater sense of self-control and self-efficacy in solving problems.

## 5. Conclusions

The results of the present study suggest that personality dimensions (sociotropy and autonomy) are associated with depressive and anxious symptoms in both MDD and GAD patients, which confirms the dual and transdiagnostic vulnerability of these dimensions. Furthermore, the examination of the mediating role of emotional regulation strategies (brooding, reflection and defenses immature) provides evidence that brooding is the essential strategy to explain the pathway along which personality dimensions can lead to depressive and anxious symptoms in patients with MDD and GAD. Our findings also show that immature defense styles function as mediators together with brooding, and they do so by linking autonomy and sociotropy with depressive symptoms in patients with MDD and with GAD, respectively. Finally, this study confirms the maladaptive role of reflection, specifically in patients with GAD when anxiety symptoms are evaluated as a dependent variable.

## Figures and Tables

**Figure 1 ijerph-17-05716-f001:**
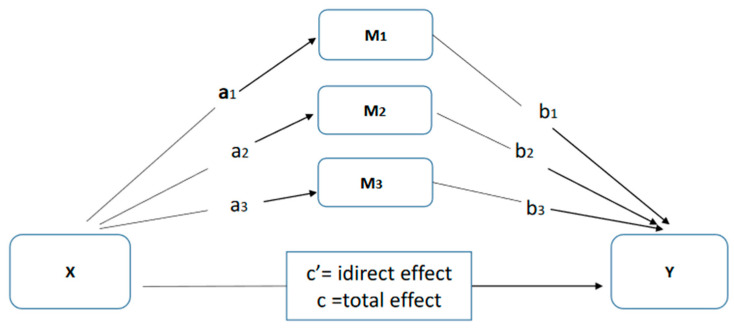
Illustration of a multiple mediation design. Path c represents the relationship between X and Y (total effect). The total effect is the sum of the direct and the indirect effect (c = c’ + a × b). Path c’ (direct effect) represents the relationship between X and Y when controlling for M (mediators).

**Table 1 ijerph-17-05716-t001:** Descriptive statistics and bivariate correlations of the Major Depressive Disorder (MDD) patients (*n* = 119) and Generalized Anxiety Disorder (GAD) patients (160).

Variables	MDD	GAD	1	2	3	4	5	6	7
	Mean (SD)	Mean (SD)							
1.Sociotropy	108.8 (15.2)	103.7 (16.0)	--	0.28 **	0.43 **	0.31 **	0.38 **	0.31 **	0.28 **
2.Autonomy	94.6 (14.5)	90.1 (14.6)	0.04	--	0.27 **	0.23 **	0.55 **	0.38 **	0.20 *
3. Brooding	15.0 (3.0)	13.9 (3.3)	0.40 **	0.26 **	--	0.40 **	0.42 **	0.48 **	0.37 **
4. Reflection	12.0 (3.0)	10.4 (2.7)	0.20 *	0.25 **	0.50 **	--	0.26 **	0.30 **	0.33 **
5. Immature defenses	94.5 (21.4)	85.7 (20.6)	0.15	0.57 **	0.34 **	0.21 *	--	0.40 **	0.24 **
6. Depressive symptoms	35.0 (10.7)	23.6 (12.3)	0.31 **	0.28 **	0.48 **	0.28 **	0.42 **	--	0.55 **
7. Anxiety symptoms	29.0 (18.8)	27.5 (13.2)	0.30 **	0.20 **	0.45 **	0.23 **	0.22 *	0.52 **	--

Notes: * *p* ≤ 0.05; ** *p* ≤ 0.001; in the lower part of the diagonal, the correlations of the MDD patients are presented, and in the upper part, the GAD patients are presented.

**Table 2 ijerph-17-05716-t002:** Results of multiple mediation analysis of the relation between sociotropy and autonomy, respectively, and depressive symptoms in patients with MDD (*n* = 119).

Mediation Pathway	Point Estimate	SE	BC 95% CI	
			Lower	Upper
**Sociotropy → Cognitive variables → Depressive symptoms 59%**				
Completely standardized indirect effects				
TOTAL	0.18	0.06	0.07	0.31
Immature defenses	0.05	0.03	−0.01	0.12
Brooding	0.12	0.06	0.02	0.24
Reflection	0.01	0.03	−0.04	0.07
Contrasts				
Immature defenses vs. brooding	0.07	0.06	−0.05	0.21
Immature defenses vs. reflection	−0.04	0.05	−0.14	0.05
Brooding vs. reflection	0.11	0.07	−0.02	0.25
**Autonomy→ Cognitive variables → Depressive symptoms 90%**				
Completely standardized Indirect effects				
TOTAL	0.28	0.07	0.15	0.41
Immature defenses	0.17	0.06	0.07	0.30
Brooding	0.10	0.04	0.02	0.19
Reflection	0.01	0.03	−0.06	0.08
Contrasts				
Immature defenses vs. brooding	−0.08	0.07	−0.23	0.07
Immature defenses vs. reflection	−0.16	0.07	−0.30	−0.03
Brooding vs. reflection	0.08	0.06	−0.03	−0.22

**Table 3 ijerph-17-05716-t003:** Results of multiple mediation analysis of the relation between sociotropy and autonomy, respectively, and depressive symptoms in patients with GAD (*n* = 160).

Mediation Pathway	Point Estimate	SE	BC 95% CI	
			Lower	Upper
**Sociotropy → Cognitive variables → Depressive symptoms 79.16%**				
Completely standardized indirect effects				
TOTAL	0.19	0.04	0.12	0.28
Immature defenses	0.06	0.03	0.02	0.13
Brooding	0.10	0.03	0.05	0.18
Reflection	0.02	0.02	−0.01	0.07
Contrasts				
Immature defenses vs. brooding	−0.04	0.05	−0.13	0.05
Immature defenses vs. reflection	0.04	0.03	−0.02	0.11
Brooding vs. reflection	0.08	0.04	0.006	0.17
**Autonomy→ Cognitive variables → Depressive symptoms 45.16%**				
Completely standardized indirect effects				
TOTAL	0.15	0.04	0.06	0.24
Immature defenses	0.06	0.04	−0.01	0.14
Brooding	0.07	0.02	0.03	0.13
Reflection	0.01	0.02	−0.01	0.06
Contrasts				
Immature defenses vs. brooding	−0.01	0.05	−0.11	0.09
Immature defenses vs. reflection	0.04	0.04	−0.04	0.13
Brooding vs. reflection	0.06	0.03	−0.001	0.12

**Table 4 ijerph-17-05716-t004:** Results of multiple mediation analysis of the relation between sociotropy and autonomy, respectively, and anxiety symptoms in patients with MDD (*n* = 119).

Mediation Pathway	Point Estimate	SE	BC 95% CI	
			Lower	Upper
**Sociotropy → Cognitive variables → Anxiety symptoms 59.2%**				
Completely standardized indirect effects				
TOTAL	0.17	0.05	0.07	0.29
Immature defenses	0.00	0.00	−0.00	0.00
Brooding	0.16	0.06	0.05	0.28
Reflection	−0.00	0.00	−0.00	0.00
Contrasts				
Immature defenses vs. brooding	0.15	0.06	0.03	0.28
Immature defenses vs. reflection	−0.01	0.03	−0.07	0.05
Brooding vs. reflection	0.16	0.07	0.03	0.30
**Autonomy→ Cognitive variables → Anxiety symptoms 54%**				
Completely standardized indirect effects				
TOTAL	0.14	0.06	0.01	0.26
Immature defenses	0.02	0.05	−0.09	0.14
Brooding	0.12	0.04	0.04	0.21
Reflection	−0.00	0.03	−0.08	0.05
Contrasts				
Immature defenses vs. brooding	0.09	0.08	−0.06	0.25
Immature defenses vs. reflection	−0.03	0.06	−0.16	0.09
Brooding vs. reflection	0.12	0.06	0.01	0.27

**Table 5 ijerph-17-05716-t005:** Results of multiple mediation analysis of the relation between sociotropy and autonomy, respectively, and anxiety symptoms in patients with GAD (*n* = 160).

Mediation Pathway	Point Estimate	SE	BC 95% CI	
			Lower	Upper
**Sociotropy → Cognitive variables → Anxiety symptoms 52%**				
Indirect effects				
TOTAL	0.13	0.04	0.06	0.23
Immature defenses	0.02	0.03	−0.02	0.08
Brooding	0.06	0.03	0.001	0.13
Reflection	0.04	0.02	0.01	0.11
Contrasts				
Immature defenses vs. brooding	−0.04	0.05	−0.13	0.05
Immature defenses vs. reflection	−0.02	0.04	−0.10	0.05
Brooding vs. reflection	0.02	0.05	−0.08	0.11
**Autonomy → Cognitive variables → Anxiety symptoms 93.3%**				
Indirect effects				
TOTAL	0.14	0.05	0.04	0.24
Immature defenses	0.04	0.05	−0.04	0.14
Brooding	0.05	0.02	0.01	0.11
Reflection	0.04	0.02	0.01	0.10
Contrasts				
Immature defenses vs. brooding	−0.01	0.06	−0.12	0.10
Immature defenses vs. reflection	0.01	0.05	−0.10	0.11
Brooding vs. reflection	0.01	0.03	−0.06	0.08

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
