# Peer review of "Sociotropy, Autonomy and Emotional Symptoms in Patients with Major Depression or Generalized Anxiety: The Mediating Role of Rumination and Immature Defenses"

_ijerph, 2020, doi:10.3390/ijerph17165716_

Round 1

Reviewer 1 Report

The paper is interesting. I suggest in results: Table 1 is confuse, I don't kwon in line 1, 2, 3.... this have clarify. And the autor need check the language.

Author Response

The paper is interesting. I suggest in results: Table 1 is confuse, I don't kwon in line 1, 2, 3.... this have clarify. And the autor need check the language.

ANSWER: We appreciate your comment. Following your suggestion, we have modified the presentation of Table 1 to make it more understandable to the reader. In order to avoid presenting two tables, Table 1 shows the statistics of the two patient samples (MDD and GAD) in the columns following the list of variables. Furthermore, the Pearson's bivariate correlations of both samples are presented. In the lower part of the diagonal, those corresponding to patients with MDD; in the upper part of the diagonal, those of patients with GAD. The numbers that appear at the top of the Table (1,2,3 ...) identify each of the analyzed variables. Please, see Table 1 in the revised document (p. 6)

Reviewer 2 Report

Perhaps what could enhance this results of this study would be more direct case study application regarding how maladaptive coping styles might be specifically alleviated or processed in a clinical setting. 

Author Response

Perhaps what could enhance this results of this study would be more direct case study application regarding how maladaptive coping styles might be specifically alleviated or processed in a clinical setting. 

ANSWER: Following your suggestion, in the revised version of the manuscript we have added new information on how to manage maladaptive coping styles in a clinical setting. Our changes to the revised manuscript are highlighted in color to make them easily identifiable (please, see p. 12, lines 333-344).

Reviewer 3 Report

This study identifies important links in the chain of events that lead to major depression or generalized anxiety disorder along specific personality dimensions.  The rational for the study is well stated, the constructs are well defined and the analytical methods are appropriate to the study aims. The only thing missing seems is a comment on the implications of these findings for mental health practitioners and/or how these findings could be used to identify at risk persons, and protect or improve mental health.

Author Response

This study identifies important links in the chain of events that lead to major depression or generalized anxiety disorder along specific personality dimensions.  The rational for the study is well stated, the constructs are well defined and the analytical methods are appropriate to the study aims. The only thing missing seems is a comment on the implications of these findings for mental health practitioners and/or how these findings could be used to identify at risk persons, and protect or improve mental health.

ANSWER: We appreciate your favorable comments on the study and your constructive suggestion. In the revised version we have added new information for mental health practitioners (please, see p. 12, lines: 331-347).

Reviewer 4 Report

This paper has potentially valuable data that investigates the potential mediation effects of rumination and 'immature defences' on the association between personality (sociotropy and autonomy) and depression and anxiety. My major reservation with the paper is how the analysis is explained (this is not clear) and conducted. Overall, I would recommend a sequence of structural equation models, or path analyses, where symptoms of depression and anxiety and modeled simultaneously. My comments are below.

Major

The format of the paper should follow STROBE guidelines (follow all headings, remove results pertaining to the sample to the Results, in a Table, stratified by groups).

MDD and GAD are highly co-morbid, and some classifications require that MDD ‘trumps’ GAD (becoming the primary diagnosis when both are present). To what extent did MDD and GAD overlap in this study? I would expect 3 groups – MDD, GAD and mixed GAD/MDD?

Analysis: You state that mediation models are used, but what is the overall anlaysis strategy (path analysis, SEM)? More detail is needed.

Results:

Table 1 - Report ‘ns’ is not acceptable. The actual r-value should be reported (without the asterisk to indicate non-singificance)

It is difficult to understand the results, as no figure is presented and the analysis is not clearly outlined. For example, I would expect to see some form of path analysis or SEM, modelling the two outcomes simultaneously (with these outcomes correlated, due to the overlap between GAD and MDD, but also the results of the correlation Table 1). The mediators should also be correlated, and be between the paths from personality factors to the outcomes. Personality factors are alos correlated? It appears that currently, mediators are being modelled singly, to individual outcomes, but this is not what is appropriate according to Table 1, and it is also reducing power by limiting your sample size?

Pairwise contrasts need to be explained? The audience for this journal will not be familiar with a lot of this terminology. Unfortunately, I cannot understand a lot of what is happening here (and I am familiar with SEM). For me, if I could see this in an overall figure/model, with the results on the paths, it would be more understandable – along with explanation in the analysis section.

Minor

“The relationship between these personality dimensions and depressive symptoms has been widely documented”

After a statement like this I would expect to see citations to a meta-anlaysis. Have there been any meta-analyses in this area? If so, these should be cited. If not, then this should be rephrased.

“For example, Spasojević and Alloy [25] informed that rumination mediated the relationship between self-criticism and the subsequent development of depression, but this result could not be replicated with dependence (self-criticism and 70 dependence are similar constructs to autonomy and sociotropy, respectively).”

How similar? Similar in a theoretical sense, or in terms of their operationalisation within scales? I am finding it difficult to see how self-criticism corresponds to autonomy, for example

“The ruminative process, though, has two different components: ‘brooding’, which refers to a passive, cyclical focus on negative emotions, and 75 ‘reflection’, referring to emotionally neutral pondering”.

Ok, but has this finding been replicated?

The measures section is confusing. If Sociotropy consists of 3 subscales, does this then map to 3 separate dimensions (which should then be modelled separately) – and then an overall alpha is inappropriate? Or does it remain as a single dimension?

Lines 140-144 – this needs to be explained more clearly. Why would you measure these constructs and then not report them?

Author Response

This paper has potentially valuable data that investigates the potential mediation effects of rumination and 'immature defenses' on the association between personality (sociotropy and autonomy) and depression and anxiety. My major reservation with the paper is how the analysis is explained (this is not clear) and conducted. Overall, I would recommend a sequence of structural equation models, or path analyses, where symptoms of depression and anxiety and modeled simultaneously. My comments are below.

ANSWER: We are very grateful for your appreciation of our work and your detailed comments. We have modified the manuscript following your suggestions. Please find our answers point by point as follows. We have provided the page numbers and lines relating to specific modifications of the manuscript.

 Major

The format of the paper should follow STROBE guidelines (follow all headings, remove results pertaining to the sample to the Results, in a Table, stratified by groups).

ANSWER: Following your suggestion, we have revised the paper format according to STROBE guidelines. Specifically, we have transferred the information belonging to the sample to the Results section. (Please, see p.5, lines: 205-209).

 MDD and GAD are highly co-morbid, and some classifications require that MDD ‘trumps’ GAD (becoming the primary diagnosis when both are present). To what extent did MDD and GAD overlap in this study? I would expect 3 groups – MDD, GAD and mixed GAD/MDD?

ANSWER: Thank you for giving us the opportunity to clarify this aspect of our work. We certainly agree with the Reviewer’s comment on the need to further explain this aspect in the manuscript.

Anxiety and depression are among the most common psychiatric disorders, and there is a high comorbidity between the two. Furthermore, there is a high prevalence of anxiety symptoms in depressive disorders and depressive symptoms in anxiety disorders that do not meet the criteria for diagnosing a fully-fledged disorder (Bakish, 1999; Nutt, 1997).

Taking the above into account, and given that our study sought to discover whether the use of coping strategies is specific or transdiagnostic in emotional disorders, patients with a pure MDD or a pure GAD were selected, using the Structured Clinical Interview for DSM- IV-TR (First, et al., 2005). (Please, see p. 3, lines: 108-116).

Analysis: You state that mediation models are used, but what is the overall anlaysis strategy (path analysis, SEM)? More detail is needed.

ANSWER: as suggested, more information on PROCESS is provided in the revised version of the manuscript. (Please, see p.5, lines: 185-200)

PROCESS is a computational tool available for SPSS that simplifies the implementation of mediation, moderation, and conditional process analysis with observed variables (Hayes, 2012, 2013; Hayes & Little, 2018). This tool has been widely studied (Hayes et al., 2017; Hayes & Little, 2018; Hayes & Rockwood, 2020; Wright & Hallquist, 2020), and used in numerous studies published in peer reviewed journals.

As it is stated by Hayes et al., (2017): “Does it matter whether one uses PROCESS as opposed to an SEM program? Specifically, will one's results be influenced by whether PROCESS or an SEM program is used? Given that SEM and PROCESS are based on different estimation methods and theory, some differences can be expected. However, for models of observed variables (i.e., nothing latent), differences in results tend to be trivial, and rarely will the substantive conclusions a researcher arrives at be influenced by the decision to use PROCESS rather than SEM”.

 Results:

Table 1 - Report ‘ns’ is not acceptable. The actual r-value should be reported (without the asterisk to indicate non-singificance)

ANSWER: As suggested, we have now included non-significant coefficients. Please, see Table 1 (p.6)

 It is difficult to understand the results, as no figure is presented and the analysis is not clearly outlined. For example, I would expect to see some form of path analysis or SEM, modelling the two outcomes simultaneously (with these outcomes correlated, due to the overlap between GAD and MDD, but also the results of the correlation Table 1). The mediators should also be correlated, and be between the paths from personality factors to the outcomes. Personality factors are alos correlated? It appears that currently, mediators are being modelled singly, to individual outcomes, but this is not what is appropriate according to Table 1, and it is also reducing power by limiting your sample size?

ANSWER: Thank you for these important recommendations and suggestions. We have included a Figure indicating the different elements of a multiple mediation model and we have also added a more detailed explanation for readers less familiar with this analysis. Please see p.5

Regarding the correlation coefficients, the three maladaptive coping strategies showed significant associations in both patient samples. However, and as expected, the two components of rumination generally reached higher values. Brooding and reflection are similar at the process level (repetitive thinking, unproductive thinking), but may differ depending on the valence (emotion-centered), the context in which it occurs, and the level of interpretation adopted (Watkins, 2008). Whereas the brooding component has consistently been related to more depression, both concurrently and longitudinally (e.g., Aldao et al., 2010; Burwell and Shirk, 2007; Merino et al., 2016; Olson and Kwon, 2008), the role of reflection in depressive symptoms has thus far not been determined (Burwell and Shirk, 2007). Overall, evidence suggests that reflective rumination is also associated with depression, but not at the same magnitude as brooding (Olatunji et al., 2013).

In relation to the personality dimensions, we obtained an interesting result. In the group of patients with MDD, both dimensions were not significantly associated; however, in the group of patients with anxiety, they showed a significant association. As noted in the Discussion section: This finding seems to suggest that the ‘modes of functioning, values, and goals’ that constitute vulnerabilities to emotional symptoms are more clearly delineated in MDD patients than in GAD patients. That is, in depressed patients the vulnerability was focused towards intrapersonal or interpersonal domain stressors; in patients with GAD, by contrast, the significant interrelationship between sociotropy and autonomy reflects the existence of a ‘shared feature’ between both types of vulnerability (p. 10, lines: 265-270).

Pairwise contrasts need to be explained? The audience for this journal will not be familiar with a lot of this terminology. Unfortunately, I cannot understand a lot of what is happening here (and I am familiar with SEM). For me, if I could see this in an overall figure/model, with the results on the paths, it would be more understandable – along with explanation in the analysis section.

ANSWER: Thank you for giving us the opportunity to clarify this aspect in the text. In multiple mediation analysis, the pairwise contrast of two indirect effects allows us to examine whether these two indirect effects differ significantly. This method can be used to test the equality of any two specific indirect effects when including multiple mediators in the model, which allows us to examine whether one indirect effect is different from another (Hayes & Little, 2018). Specifically, when the pairwise confidence interval includes zero, this indicates that the two indirect effects cannot be distinguished in terms of magnitude.

As suggested, in the analysis section of the revised version this method is clarified to make the analysis more easily understandable. In addition, a graph illustrating a general multiple mediation model is now included. Please, see p. 5, Figure 1.

Minor 

“The relationship between these personality dimensions and depressive symptoms has been widely documented”

After a statement like this I would expect to see citations to a meta-anlaysis. Have there been any meta-analyses in this area? If so, these should be cited. If not, then this should be rephrased.

ANSWER: We appreciate this comment. The previous literature does not allow us to say that this result has been extensively documented, therefore we have rewritten this sentence more precisely (please, see p. 2, line 46).

 “For example, Spasojević and Alloy [25] informed that rumination mediated the relationship between self-criticism and the subsequent development of depression, but this result could not be replicated with dependence (self-criticism and 70 dependence are similar constructs to autonomy and sociotropy, respectively).”

How similar? Similar in a theoretical sense, or in terms of their operationalisation within scales? I am finding it difficult to see how self-criticism corresponds to autonomy, for example

ANSWER: We certainly agree with the Reviewer’s comment. This aspect is clarified in the revised version of the manuscript. They are theoretically and operationally similar constructs. In terms of vulnerability stress, psychodynamic (Blatt, 1974; Bowlby, 1980; and Arieti & Bemporad, 1980) and cognitive (Beck, 1983; Clark et al., 1999) postulates have many concomitants to explain what dimensions of personality can make people vulnerable to developing depression. In this sense, dependency and sociotropy have in common an exaggerated opinion of others and an overvaluation of interpersonal relationships. Self-criticism and autonomy share a desire for independence, motivation for achievement, and fear of failure, which makes these individuals tend to self-criticize and blame themselves (please, see p.2, line 74).

 “The ruminative process, though, has two different components: ‘brooding’, which refers to a passive, cyclical focus on negative emotions, and 75 ‘reflection’, referring to emotionally neutral pondering”.

Ok, but has this finding been replicated?

ANSWER: Effectively, these two components of rumination have been empirically established in both adolescents and adults (Burwell & Shirk, 2007; Treynor et al., 2003). (Please see p.2, lines: 78-79).

 The measures section is confusing. If Sociotropy consists of 3 subscales, does this then map to 3 separate dimensions (which should then be modelled separately) – and then an overall alpha is inappropriate? Or does it remain as a single dimension? (please, see p.4, lines: 140-142).

ANSWER: The Sociotropy dimension includes three different facets that define this personality trait. Similarly, the autonomy dimension also encompasses three different characteristics. Generally, as in the present study, the global dimension measure is used, since it provides a more complete assessment. Internal consistency values ​​for both dimensions are provided.

Lines 140-144 – this needs to be explained more clearly. Why would you measure these constructs and then not report them?

ANSWER: We agree with the reviewer's comment. This study really is part of a larger investigation using the full DSQ-40. The current wording, however, might be confusing, and hence the new version of the manuscript was changed to clarify that only immature defenses were used; in addition, the reasons why this choice was made continue to be provided. (Please, see p. 4, lines:157-159).

Round 2

Reviewer 4 Report

This manuscript has improved from the last submission. Now that that analysis is explained more clearly, however, I still have some concerns. These are as follows:

Major

  • The big issue is why the authors choose to model autonomy and sociotropy separately, rather than simultaneously? This is important as a) they are correlated for GAD, and b) you are running more statistical tests than necessary, increasing likelihood of error. Perhaps this PROCESS feature can only model these variables singly (if this is the case, then this should be added as a major limitation)? If so, then why not consider SEM, where they could be modelled simultaneously?
  • Discussion: “In this sense, the present study revealed that brooding is the link between personality vulnerabilities and depressive symptoms in patients with MDD and with GAD, thus demonstrating its transdiagnostic function.” I don’t think the results suggest this – brooding may be A link, not necessarily THE link – this should be toned down? Especially given that brooding was not consistently significantly different to other mediators in the pairwise contrasts analyses? Also,as per the above comment,  the authors have chosen to model the traits separately, leading to multiple tests and the possibility of chance findings
  • Author contributions: The author initials in this section do not correspond with the listed authors? E.g. EB is listed in the author contributions, but not listed as an author, etc.? 

Minor

  • There is an error in Fig 1 – ‘idirect’
  • “A total of four models were tested for each group of patients.” – what were these 4 models - describe them here?
  • Results – patient demographics should be presented as a Table, as per STROBE

Author Response

This manuscript has improved from the last submission. Now that that analysis is explained more clearly, however, I still have some concerns. These are as follows:

ANSWER: We sincerely thank you for your appreciation of our work and you’re your detailed and constructive comments; we have amended the manuscript following your suggestions.

Major

  • The big issue is why the authors choose to model autonomy and sociotropy separately, rather than simultaneously? This is important as a) they are correlated for GAD, and b) you are running more statistical tests than necessary, increasing likelihood of error. Perhaps this PROCESS feature can only model these variables singly (if this is the case, then this should be added as a major limitation)? If so, then why not consider SEM, where they could be modelled simultaneously?

ANSWER: Thank you for giving us the opportunity to clarify this aspect of our work.

Psychodynamic and cognitive formulations of depression in terms of personality event congruence hypotheses have been postulated (Beck, 1983, Blatt and Zuroff, 1992; Hammen et al., 1985). According to these theorists, there are two different dimensions of personality that make people particularly vulnerable to developing depression when faced with certain events consistent with their predisposition. Specifically, they point out that sociotropic individuals are at increased risk of developing depression after negative interpersonal events (loss, rejection, or conflict), while autonomous individuals would be particularly vulnerable to depression after negative achievement events (low performance assessment, limitations on self-determination, decreased control). Studies that have examined the vulnerability-stress hypothesis have obtained mixed results. Thus, although there is consistent evidence regarding sociotropic vulnerability, the findings regarding the autonomy dimension are inconsistent (Bakhshani et al., 2007; Connor-Smith et al., 2002; Masih et al., 2007).

Furthermore, although these personality dimensions were originally formulated as risk factors for depression, scholars have wondered if they might also confer risk for anxiety disorders (e.g., Alloy et al., 1990; Zuroff et al., 2004), among other reasons, due to the overlap between depressive and anxious symptoms. Again, the results in this area are inconclusive (Alford and Gerrity, 1995; Fresco et al., 2001).

In our opinion, the reasons mentioned above support the independent analysis of sociotropy and autonomy (the first aim of our study), because in this way we can know:

- whether both dimensions equally predispose someone to depression and

- whether these dimensions are specific to depressive disorders or may also be factors of vulnerability to anxiety.

On the other hand, the literature highlights the need to investigate the relationship between personality and emotional psychopathology, analyzing more than one diagnostic category and more than one emotional regulation strategy as an intermediate link between both variables (D'Avanzato et al., 2013). Addressing this gap was indeed the motivation for the second aim of our study.

The evidence consistently supports a high comorbidity between MDD and GAD, not only at the diagnostic level but also the symptomatic level. The inclusion in this study of patients diagnosed with pure MDD or pure GAD allows us to see whether the use of emotional regulation strategies is specific or transdiagnostic in emotional disorders. The inclusion in this study of patients diagnosed with pure MDD or pure GAD allows us to see whether the use of emotional regulation strategies is specific or transdiagnostic in emotional disorders. In this vein, we examined: a) specific trajectories of depression and anxiety, that is, depressive symptoms in patients with MDD, and anxiety symptoms in patients with GAD, and, b) comorbid trajectories, that is, anxiety symptoms in patients with MDD, and depressive symptoms in patients with GAD. To carry out these aims, the PROCES macro developed by Hayes (2018) is a very suitable tool, since it allows for an examination of the simultaneous contribution of multiple mediators as well as the particular contribution of each one.

  • Discussion: “In this sense, the present study revealed that brooding is the link between personality vulnerabilities and depressive symptoms in patients with MDD and with GAD, thus demonstrating its transdiagnostic function.” I don’t think the results suggest this – brooding may be A link, not necessarily THE link – this should be toned down? Especially given that brooding was not consistently significantly different to other mediators in the pairwise contrasts analyses? Also,as per the above comment,  the authors have chosen to model the traits separately, leading to multiple tests and the possibility of chance findings.

ANSWER: We agree. As recommended, we have now nuanced the role of brooding according to pairwise contrasts.

In this sense, the present study revealed that brooding mediates the relationship between personality vulnerabilities and depressive symptoms in patients with MDD and GAD, thus demonstrating its transdiagnostic function. However, as the pairwise contrasts indicate, the effect of the other two mediators, particularly immature defenses, cannot be ruled out. (Please, see p.12, lines: 283-285).

  • Author contributions: The author initials in this section do not correspond with the listed authors? E.g. EB is listed in the author contributions, but not listed as an author, etc.? 

ANSWER: We would like to apologize for this mistake. The contribution of each of the authors has been correctly indicated in the new version.

Minor

  • There is an error in Fig 1 – ‘idirect’

ANSWER: We apologize for this oversight. In the new version it has been corrected (p. 5)

  • “A total of four models were tested for each group of patients.” – what were these 4 models - describe them here?

ANSWER: We certainly agree with the Reviewer’s comment on the need to further explain this aspect in the manuscript. We have added the following explanation in the manuscript: A total of four models were tested for each group of patients. That is, the role of the three mediators in the relationship between each of the personality dimensions (sociotropy and autonomy) and the depressive and anxious symptoms, respectively, was analyzed. (Please, see p. 5, lines:

  • Results – patient demographics should be presented as a Table, as per STROBE

ANSWER: As recommended, in the revised version of the manuscript we have added a Table with the demographic characteristics of each group of patients. Please, see Table 1 (p. 6).

References:

Alloy, L. B., Kelly, K. A., Mineka, S., & Clements, C. M. (1990). Comorbidity of anxiety and depressive disorders: A helplessness-hopelessness perspective. In J. D. Maser & C. R. Cloninger (Eds.), Comorbidity of mood and anxiety disorders (pp. 499–543). Washington, DC: American Psychiatric Press.

Alford, B.A.; Gerrity, D.M. The Specificity of Sociotropy‐autonomy Personality Dimensions to Depression Vs. Anxiety. J. Clin. Psychol. 1995, 51, 190-195.

Bakhshani, N.M. Role of Personality Styles (Sociotropy/Autonomy) and Moderating Effects of Social Support in Clinically Depressed Patients. J. Med. Sci. 2007, 7, 106-110.

Beck, A.T. Cognitive Therapy of Depression: New Perspectives. In Treatment of depression: Old controversies and new approaches; Clayton, P., Barrett, J., Eds.; Raven Press: New York, NY, USA, 1983; pp. 265-290.

Blatt, S. J., & Zuroff, D. C. (1992). Interpersonal relatedness and selfdefinition: Two prototypes for depression. Clinical Psychology Review, 12, 527-562.

Connor-Smith, J.K.; Compas, B.E. Vulnerability to Social Stress: Coping as a Mediator or Moderator of Sociotropy and Symptoms of Anxiety and Depression. Cognit. Ther. Res. 2002, 26, 39-55.

D’Avanzato, C.; Joormann, J.; Siemer, M.; Gotlib, I.H. Emotion Regulation in Depression and Anxiety: Examining Diagnostic Specificity and Stability of Strategy Use. Cognit. Ther. Res. 2013, 37, 968-980.

Fresco, D.M.; Sampson, W.S.; Craighead, L.W.; Koons, A.N. The Relationship of Sociotropy and Autonomy to Symptoms of Depression and Anxiety. J. Cogn. Psychother. 2001, 15, 17-31.

Hammen, C., Marks, T., Mayol, A., & deMayo, R. (1985). Depressive self-schematas, life-stress, and vulnerability to depression. Journal of Abnormal Psychology, 94, 308-319.

Masih, S.; Spence, S.H.; Oei, T.P. Sociotropic and Autonomous Personality and Stressful Life Events as Predictors of Depressive Symptoms in the Postpartum Period. Cognit. Ther. Res.  2007, 31, 483-502.

Mineka, S., Pury, C. L., & Luten, A. G. (1995). Explanatory style in anxiety and depression. In G. M. Buchanan & M. E. P. Seligman (Eds.), Explanatory style (pp. 135–158). Hillsdale, NJ: Lawrence Erlbaum Associates, Inc.

Şahin, N.; Ulusoy, M.; Şahin, N. Exploring the Sociotropy‐autonomy Dimensions in a Sample of Turkish Psychiatric Inpatients. J. Clin. Psychol. 2003, 59, 1055-1068.